# FAST AND SCALABLE INVERSION OF CONVOLUTION LAYERS

## ABSTRACT

Data inversion in neural networks allows to map intermediate network variables to their input source. Inversion of convolutional layers is not straightforward and is often performed approximately by training additional inversion networks. Approaching this as a linear operator inversion problem requires extremely large computational and memory resources, as huge matrices are involved. In this work we present *Scalable TRimmed Iterative Projections* (STRIP), a fast and sparse linear solver dedicated to the convolutional inversion problem.

We take advantage of the neural convolution structure to design a series of very fast projections (following the *block Kaczmarz* paradigm). We prove conditions for convergence for the two-strip case and propose a measure to estimate the rate of error reduction for the general case. In practice, we show that a single pass over the inversion matrix by STRIP can almost perfectly solve the inversion problem. Our algorithm is fast, low on memory and can scale to very large matrices. We do not have to store the linear matrix to be inverted, hence can surpass by 3 orders of magnitude linear sparse solvers, such as conjugate gradient. Extensive experiments demonstrate that our method considerably outperforms the best competing solvers by both speed and memory footprint. We further show that a single STRIP iteration is more accurate than transposed convolutions, motivating the use of such methods in U-Net architectures.

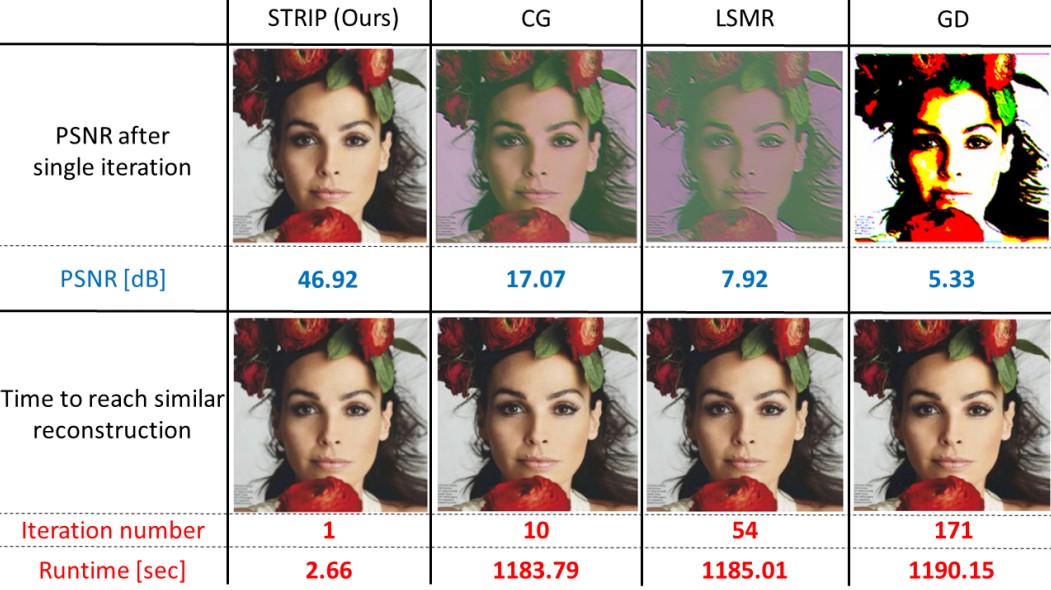

|  | STRIP (Ours) | CG | LSMR | GD |
|---|---|---|---|---|
| PSNR after single iteration | | | | |
| PSNR [dB] | 46.92 | 17.07 | 7.92 | 5.33 |
| Time to reach similar reconstruction | | | | |
| Iteration number | 1 | 10 | 54 | 171 |
| Runtime [sec] | 2.66 | 1183.79 | 1185.01 | 1190.15 |

Figure 1: Comparison of our proposed method (STRIP) against classical sparse iterative solvers on the CelebA-HQ dataset ($256^2$ pixels), for a 16 output channels convolutional layer. The top row shows reconstructions after a single iteration, with corresponding PSNR values relative to the ground truth image. The bottom row depicts reconstructions at the first iteration where each baseline surpasses the PSNR of STRIP after one iteration, along with the required iteration count and runtime.

# 1 INTRODUCTION

Convolutional layer inversion is a highly relevant research direction due to its close connection with widely used encoder-decoder architectures such as U-Net (Ronneberger et al., 2015), which play a central role in diffusion models during the reverse process. In these architectures, the encoder branch primarily employs convolutional layers to extract informative image features, while the decoder aims to accurately reconstruct the original image from these representations. Achieving reliable image reconstruction therefore requires the ability to effectively invert the operation of convolutional layers.

However, in most cases the encoder-decoder pipeline is trained jointly in an end-to-end manner. That is, most existing inversion techniques rely on learnable models, often overlooking the inherent mathematical properties of convolutional operations.

Our approach, a variant of the *block Kaczmarz* family of methods (Elfving, 1980; Needell & Tropp, 2014), formulates convolutional inversion as a linear system. This perspective enables us to directly exploit the structural attributes of the convolutional layer's matrix. By leveraging this structure, we design a fast-converging iterative inversion algorithm that achieves both efficiency and reliable reconstruction.

Our work provides mathematically grounded insights and establishes feasible conditions for fast convergence to the analytic solution, which is the *pseudoinverse* (PINV).

Our main contributions are:

1. We tackle the problem of convolutional layer inversion in a non-learning setting, relying on linear algebra fundamentals to derive the solution.

2. Unlike existing *block Kaczmarz* algorithms, our method is tailored specifically to convolutional layer matrices, enabling us to utilize their structure for an informed strip selection strategy that accelerates convergence.

3. Our algorithm avoids constructing the full or even sparse representation of the convolutional matrix, which becomes highly expensive in high dimensional settings - unlike many standard iterative solvers.

4. We provide a mathematical justification for our strip based partitioning scheme and explain the reasoning underlying our design choices.

# 2 RELATED WORK

Neural network inversion has become an active and rapidly evolving area of research. Much of the recent work focuses on interpretability, aiming to better understand the decision-making process of neural networks and to reveal their invariances, weaknesses, and blind spots (Rathjens et al., 2024; Suhail & Sethi, 2024; Zeiler & Fergus, 2014; Fel et al., 2023).

Another prominent line of work addresses *deconvolution* for tasks such as denoising and deblurring. It is important to note, however, that the operation of a convolution layer in a neural network differs from classical convolution. In the classical case, convolution involves flipping the kernel before computing the weighted sum, while in neural networks the operation corresponds to *cross-correlation*. More importantly, neural network convolution layers operate simultaneously across multiple input channels, whereas classical convolution typically applies to a single channel (or separately across channels in the multi-channel setting). This distinction explains why classical approaches such as FFT-based deconvolution (Cooley & Tukey, 1965) or Wiener filtering (Wiener, 1949) are not directly applicable to convolution layer inversion in neural networks. These methods assume the properties of classical convolution and therefore fail to capture the multi-channel structure inherent in modern convolutional layers (for further discussion, see Appendix A.1).

Many inversion approaches rely on learnable models, including neural networks and more recently diffusion-based methods, to approximate the mapping from outputs back to inputs. While these approaches have achieved impressive results, they often lack the mathematical grounding that leverages the forward mapping (input $\rightarrow$ output) to explicitly construct the inverse mapping (output $\rightarrow$ input). As a result, they are less adaptive to specific scenarios and entail considerable computational cost during both training and inference.

A widely adopted strategy for implementing convolutional layer inversion in a learnable framework is the use of *transposed convolution* (or up-convolution) layers (Long et al., 2015). This operation is most commonly employed in decoder branches of architectures such as U-Net (Ronneberger et al., 2015), where it serves to progressively reconstruct spatial structure from compressed representations. Transposed convolutions are also frequently used as standalone inversion modules for intermediate feature maps in neural networks (Dosovitskiy et al., 2016). These methods can produce strong reconstruction results. However, they do not constitute a true mathematical inversion. Moreover, they require a costly training phase prior to deployment, which further limits their efficiency and adaptability.

When a convolutional layer is viewed as a linear system, one can leverage classical linear solvers: either by analytically inverting the corresponding convolution matrix, or by computing approximate solutions through iterative methods, some more traditional including *conjugate gradient* (Hestenes et al., 1952), *Gauss Seidel* (Seidel, 1873), *gradient descent* (Cauchy et al., 1847), *LSMR* (Fong & Saunders, 2011) or *LSQR* (Paige & Saunders, 1982), and some less familiar such as *block Kaczmarz* methods (Elfving, 1980) (more details can be found in Section 3.2). The shortcoming of the former lies in its massive memory and time requirements, while the latter often suffers from low accuracy in the early iterations and may demand impractically long runtimes to converge. A key limitation of the linear system perspective is that it overlooks the specific structure of convolutional layer matrices.

# 3 PRELIMINARIES

## 3.1 SETTING AND NOTATIONS

We focus on the problem of inverting a convolutional layer in a neural network. A natural way to formulate this inversion is as a linear system of equations. By explicitly constructing the matrix representation of the convolutional layer, we can regard it as a matrix $A$, leading to the system:

$$Ax = b, \quad A \in \mathbb{R}^{m \times n},\ x \in \mathbb{R}^n,\ b \in \mathbb{R}^m. \tag{1}$$

The setting of interest in our work is the *overdetermined* regime $(m > n)$, where the system Eq. (1) has typically 0 solutions. The solution $x^*$ we strive to find is a solution of the least-squares optimization problem:

$$x^* = argmin_x \|Ax - b\|_2 = A^\dagger b, \tag{2}$$

where $A^\dagger$ denotes the Moore-Penrose (MP) inverse (or pseudoinverse, denoted as PINV) of $A$. For an invertible $A^T A$ we have a closed form expression:

$$A^\dagger = (A^T A)^{-1} A^T. \tag{3}$$

When the system is *underdetermined* ($\infty$ solutions), for an invertible $AA^T$ we get:

$$A^\dagger = A^T (AA^T)^{-1}. \tag{4}$$

This scenario frequently arises in practical neural networks where an input is mapped to a higher number of channels to form richer feature maps. This situation is particularly relevant in architectures such as U-Net, where a more efficient alternative to learned transposed convolution layers is desirable for the decoding stage.

We can also define through the PINV the space of solutions by using the matrix kernel. Let the null-space of $A$ be

$$\mathcal{N}(A) := \{x \in \mathbb{R}^n \mid Ax = \mathbf{0}\}, \tag{5}$$

where $\mathbf{0}$ is the zero vector of length $n$. We define the following kernel matrix $K$,

$$K = I - A^\dagger A, \tag{6}$$

where $I$ is the identity matrix. Then $K$ is the orthogonal projection onto the null-space of $A$.

**Active Columns:** Let $A_i \in \mathbb{R}^{m_i \times n}$ be a submatrix of $A$, that is obtained by selecting $m_i$ rows of $A$. The set of active columns of $A_i$ is those columns that contain at least one nonzero entry. The *trimmed matrix*, denoted by $\tilde{A}_i$, consists of only the active columns of $A_i$.

**Idempotency:** Let $A$ be a square matrix. This matrix is *idempotent* if $A^2 = A$. The eigenvalues of this kind of matrix are either zero or one.

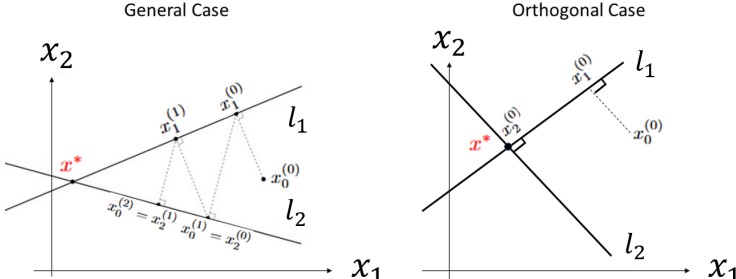

Figure 2: Illustration of our method on a two-dimensional toy problem. In the general case (left), the iterations gradually approach the solution $x^*$, requiring multiple updates for convergence. In contrast, in the orthogonal case (right), a single full sweep over the matrix is sufficient to reach the solution. The notation follows the update rules given in Eq. (9) and Eq. (10).

## 3.2 BLOCK KACZMARZ

A central challenge in higher dimensions is that explicitly constructing the full matrix $A$ is very expensive in terms of memory. To address this, iterative methods have been proposed, among which the *block Kaczmarz* (BK) algorithm is especially appealing. Instead of inverting the full matrix, BK operates by selecting one block of rows of $A$ at each iteration, solving the corresponding subproblem. The method is closely related to alternating projections (von Neumann, 1933; Bregman, 1965), where the original problem is decomposed into smaller subproblems - each corresponding to a subspace defined by a block of $A$ - where a solution is reached by iterative projections across subspaces.

Formally, for the system in Eq. (1), let $A_{i_t} \in \mathbb{R}^{m_{i_t} \times n}$ denote a block of rows of $A$ that is selected for the $t$'th iteration and $b_{i_t} \in \mathbb{R}^{m_{i_t}}$ the respective block of the right-hand side. The index $i$ is a partition of $A$, and the subindex $t$ indicates iteration. In the general case, the partition can change each iteration. At iteration $t$, the BK update is given by

$$x^{(t+1)} = x^{(t)} + A_{i_t}^{\dagger} \left( b_{i_t} - A_{i_t} x^{(t)} \right), \tag{7}$$

where $A_{i_t}^{\dagger}$ denotes the PINV of the block $A_{i_t}$.

## 4 METHOD

Our method decomposes the convolutional system into strips and iteratively integrates their solutions to approximate the PINV. Unlike the general-purpose BK algorithm, our variant exploits the convolutional structure to achieve major gains: (i) compute the PINV only in active columns and update the corresponding entries of $x$; (ii) select maximally orthogonal strips to obtain a near-idempotent block operator $\mathcal{K}$, accelerating convergence; and (iii) align strip lengths with the cyclic structure so that all strips share active columns, allowing reuse of PINV - especially simple in the *valid* convolution case.

Consider a convolution layer with kernel $k \in \mathbb{R}^{C_{\text{out}} \times C_{\text{in}} \times H_k \times W_k}$, where $C_{\text{out}}$ is the number of filters (output channels), each of dimension $\mathbb{R}^{C_{\text{in}} \times H_k \times W_k}$. The kernel has stride and padding; in our case we set *padding* $= 0$ for a *valid* convolution.

The kernel operates on a batch of input images $X \in \mathbb{R}^{B \times C_{\text{in}} \times H_{\text{in}} \times W_{\text{in}}}$, which we reshape into row-major vectors $x \in \mathbb{R}^{C_{\text{in}} H_{\text{in}} W_{\text{in}} \times B}$. The corresponding output is $b \in \mathbb{R}^{C_{\text{out}} H_{\text{out}} W_{\text{out}} \times B}$. Defining $n := C_{\text{in}} H_{\text{in}} W_{\text{in}}$, $m := C_{\text{out}} H_{\text{out}} W_{\text{out}}$, we have $x \in \mathbb{R}^{n \times B}$ and $b \in \mathbb{R}^{m \times B}$.

We denote by $A \in \mathbb{R}^{m \times n}$ the matrix representation of the convolution layer, so that applying the kernel $k$ to $X$ is equivalent to the multiplication $Ax$. The matrix $A$ is formed by concatenating $C_{\text{out}}$ blocks of identical shape (see Figure 7 in Appendix A.2). The number of rows in each block is $r = \left\lfloor \frac{H_{\text{in}} + 2 \cdot \text{padding} - H_k}{\text{stride}} \right\rfloor + 1 \times \left\lfloor \frac{W_{\text{in}} + 2 \cdot \text{padding} - W_k}{\text{stride}} \right\rfloor + 1$. A sketch of the convolution layer operation as a linear equation system is shown in Figure 7.

In the BK method we have large degrees of freedom in selecting the sub-linear systems to be solved iteratively. For general matrices, random selection is often used. However, in our case we

---

**Algorithm 1** STRIP - Scalable TRimmed Iterative Projections

---

**Input:** $h_b$ selection which determines $S$, kernel $k$, number of iterations $T$

1: Initialize: $x_0^{(0)} = 0$
2: **for** $t = 0, 1, \ldots, T$ **do**
3:      **for** $s = 1, 2, \ldots, S$ **do**
4:          $r_s \leftarrow select\_rows(kernel = k, start\_idx = s * h_b, num\_rows = h_b)$
5:          $A_s \leftarrow extract\_conv\_mat\_rows(kernel = k, selected\_rows = r_s)$
6:          $c_s \leftarrow$ columns of $A_s$ with nonzero elements - "active columns"
7:          **if** $s = 0$ and $t = 0$ **then**
8:              $\tilde{A} \leftarrow A_s[:, c_s]$
9:              $\tilde{A}^\dagger \leftarrow \texttt{pinv}(\tilde{A})$
10:            $\tilde{K} \leftarrow I - \tilde{A}^\dagger \tilde{A}$
11:          **end if**
12:          $b_s \leftarrow b[r_s, :]$
13:          $x_s^{(t)} \leftarrow x^{(t)}[:, c_s]$
14:          $x_s^{(t)} \leftarrow \tilde{K} x_{s-1}^{(t)} + \tilde{A}^\dagger b_s$
15:      **end for**
16: **end for**

**Output:** Approximate solution $x^{(T)}$

---

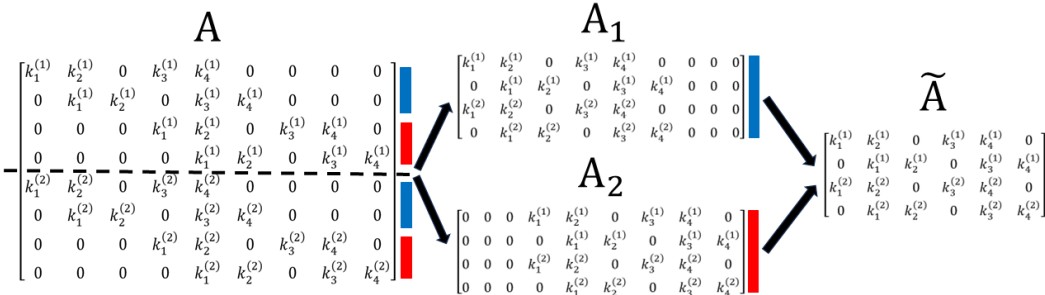

Figure 3: **Matrix trimming.** After rows are selected for the strip, most of the columns are trimmed, producing a tiny *trimmed matrix* $\tilde{A}$. The active columns of both blue and red strips contain the same elements, both producing $\tilde{A}$. *This allows to compute the PINV only once for all strips!*

show performance greatly improves by a methodological deterministic selection. These are the considerations and solutions we propose:

1. **Inter-block orthogonality:** Preserve orthogonality by selecting the same contiguous row indices from each block. Partition $A$ into $S$ equal-sized strips, each formed from identical row ranges across all blocks (Figure 5, STRIP division).

2. **Equal-sized strips:** Ensure the strip size evenly divides $r$. For each strip, choose a row count per block $h_b$ such that $h_b \mid r$, giving total strip height of $h := h_b \cdot C_{\text{out}}$.

3. **Alignment of active columns:** To guarantee common active columns for all strips, use *valid* convolution and select rows proportional to row-major shifts (Figure 3). This requires $h_b$ to divide or multiply $l := \left\lfloor \frac{W_{\text{in}} + 2 \cdot \text{padding} - W_k}{\text{stride}} \right\rfloor + 1$, while still dividing $r$.

4. **Computational efficiency:** For each strip, row indices $r_s$ and nonzero columns $c_s$ are extracted. Since the trimmed matrix $\tilde{A}$ is identical across strips (Section 3.1), we compute $\tilde{A}, \tilde{A}^\dagger$ (Eq. (4)), and $\tilde{K}$ (Eq. (6)) once and reuse them, enabling efficient updates of $x$.

The PINV step in row 9 of Algorithm 1 can be computed using several techniques. We tested multiple approaches and observed similar results, with a slight advantage for QR decomposition, which was selected as default.

## 5 MATHEMATICAL ANALYSIS

Our method follows the general BK framework, but introduces a fixed partitioning of the matrix $A$ into non-overlapping strips that is reused at every iteration. Consequently, the iteration index $t$ and the partition index $i$ are decoupled, unlike in the original formulation of the algorithm. Following this decoupling, we show a faster convergence rate in our method. To examine this rate, we use the criterion of discrete-time dynamical systems.

The BK iterative method (Eq. (7)), in our perspective, can be reformulated as

$$x_i^{(t+1)} = \left( I - A_i^\dagger A_i \right) x_i^{(t)} + A_i^\dagger b_i, \tag{8}$$

where the original equation system in Eq. (1) is split into $S$ subsystems. Then, we apply Eq. (7) to these $S$ subsystems sequentially, with initial condition $x_0^{(0)}$, to obtain:

$$
\begin{aligned}
x_1^{(t)} &= \left( I - A_1^\dagger A_1 \right) x_0^{(t)} + A_1^\dagger b_1 \\
x_2^{(t)} &= \left( I - A_2^\dagger A_2 \right) x_1^{(t)} + A_2^\dagger b_2 \\
&\vdots \\
x_S^{(t)} &= \left( I - A_S^\dagger A_S \right) x_{S-1}^{(t)} + A_S^\dagger b_S.
\end{aligned}
\tag{9}
$$

Now, for the next round, we update the next initial condition as

$$x_0^{(t+1)} = x_S^{(t)}. \tag{10}$$

Let the SVD of a matrix $A$ be $U\Sigma V^T$, where $U \in \mathbb{R}^{m \times r}$ and $V \in \mathbb{R}^{n \times r}$ have orthonormal columns, and $\Sigma \in \mathbb{R}^{r \times r}$ contains the nonnegative singular values of $A$. By applying SVD, and the dagger operator under the assumption that every subsystem is *underdetermined* (Eq. (4)), the equation system in Eq. (9) becomes

$$
\begin{aligned}
x_1^{(t)} &= \left( I - V_1 V_1^T \right) x_0^{(t)} + V_1 \hat{b}_1 \\
&\vdots \\
x_S^{(t)} &= \left( I - V_S V_S^T \right) x_{S-1}^{(t)} + V_S \hat{b}_S,
\end{aligned}
\tag{11}
$$

where $\hat{b}_i = \Sigma_i^{-1} U_i^{-1} b_i$. For justification see Eq. (17). The update rule is the same (Eq. (10)). From Eq. (11) and Eq. (10), one can formulate the relation of $x_0^{(t)}$ to $x_0^{(t+1)}$ as

$$x_0^{(t+1)} = \mathcal{K} x_0^{(t)} + \mathcal{B} \tag{12}$$

where

$$\mathcal{K} = \prod_{i=1}^{S} \left( I - V_i V_i^T \right), \quad \mathcal{B} = \sum_{j=1}^{S} \prod_{i=j+1}^{S} \left( I - V_i V_i^T \right) V_j \hat{b}_j. \tag{13}$$

The relation between successive iterations in Eq. (12), which is obtained from BK recurrence, has the form of a discrete time linear dynamical system. The merits of formulating the recurrence relation as such are 1) formulating the explicit solution of this recurrence relation and 2) applying the well-known convergence criteria. In what follows, we discuss the solution and conditions for convergence.

### 5.1 KACZMARZ RECURRENCE RELATION AS DYNAMICAL SYSTEM

BK recurrence relation in Eq. (12) has the form of a linear difference equation with constant constraint. Given an initial condition $x_0^{(0)}$, the solution to is

$$x_0^{(t)} = \mathcal{K}^t x_0^{(0)} + \left[ I + \mathcal{K} + \cdots + \mathcal{K}^{t-1} \right] \mathcal{B}. \tag{14}$$

The convergence of this solution depends on the spectrum of $\mathcal{K}$ and on the respective eigenvectors. The structure of the matrix $\mathcal{K}$, under some conditions, implies idempotency. Let us define a distance of the matrix $\mathcal{K}$ from idempotency as

**Definition 1** (Distance from Idempotency). The distance from idempotency is

$$\left\| \mathcal{K} - \mathcal{K}^2 \right\|_F \tag{15}$$

where $\mathcal{K}$ is defined as in Eq. (13).

Let us denote the column space of $V_i$ by $\mathcal{V}_i$. If the subspaces $\{\mathcal{V}_i\}_{i=1}^S$ are orthogonal, the matrix $\mathcal{K}$ would be equal to $\mathcal{K}^\perp = I - \sum_{i=1}^S V_i V_i^T$. The other (correlated) addends are for the dependency between these subspaces. The matrix $\mathcal{K}^\perp$ can be interpreted as the "idempotent" part of $\mathcal{K}$. The conditions for convergence are discussed in Appendix E. Our main results are summarized as follows:

**Theorem 2.** *A linear system as expressed by Eq. (12) and Eq. (13) admits the following:*

1. *The solution converges in one step if $\mathcal{K}$ is idempotent and $\mathcal{B}$ belongs to its kernel.*

2. *If $\mathcal{V}_i \perp \mathcal{V}_j$ for all $i \neq j$, Eq. (14) converges in one iteration.*

3. *For $S = 2$, if $I_c < 1$ then Eq. (14) converges, and if $I_d > 1$ then Eq. (14) diverges, where the indicators $I_c$ and $I_d$ are defined in Eq. (35) in Appendix E.4.*

See proofs in Appendix E.4. Our experiments indicate that the idempotency of $\mathcal{K}$ is a reliable criterion for strip selection, as shown in Section 6.1.

## 6 EXPERIMENTAL RESULTS

All experiments were conducted on an NVIDIA GeForce RTX 3090 GPU. To ensure fair time comparisons, we adapted all competing algorithms to the PyTorch framework. For our inversion experiments, we employed the *ConvNet* architecture. See full architecture descriptions and additional details in Appendix B.

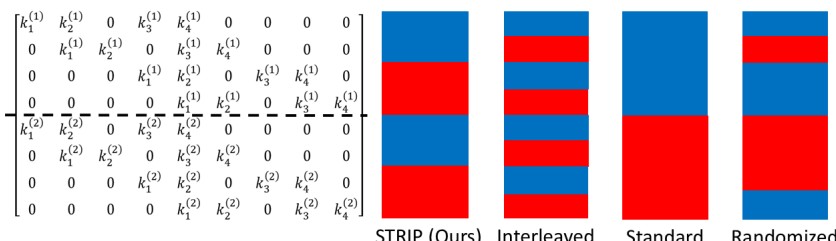

Figure 4: Illustration of different strip arrangements for partitioning a convolution matrix with two output channels (two kernels) into two strips. Rows assigned to each strip are highlighted in blue and red, respectively.

|  | STRIP (Ours) | Interleaved | Standard | Randomized |
|---|---|---|---|---|
| reconstructions | | | | |
| PSNR [dB] | 36.45 | 10.05 | 7.63 | 7.12 |
| Idempotency | 0.326 | 24.258 | 30.416 | 31.689 |

Figure 5: Idempotency of strip arrangements measured by $\|\mathcal{K} - \mathcal{K}^2\|_F$ (lower is better) on CelebA dataset. Smaller values indicate greater inter-strip orthogonality and faster convergence.

**Procedure.** Each experiment proceeds as follows: 1. Forward a batch of images through the convolution layers. 2. Add noise at a fraction of each image's standard deviation. 3. Apply the

Table 1: Comparison of STRIP versus iterative methods across datasets. All methods are evaluated after a single iteration. Best in **bold**.

| | | STRIP (Ours) | CG | GS | GD | LSQR | LSMR | RSHK |
|---|---|---|---|---|---|---|---|---|
| **MSE** ↓ | MNIST | **0** | 0.468 | 0.256 | 0.907 | 0.746 | 0.795 | 0.406 |
| | CelebA | **0** | $0.325 \pm 0.008$ | $0.273 \pm 0.007$ | $0.360 \pm 0.007$ | $0.353 \pm 0.008$ | $0.356 \pm 0.008$ | $0.313 \pm 0.008$ |
| | CelebA-HQ | **0** | $0.312 \pm 0.006$ | – | $0.347 \pm 0.005$ | $0.336 \pm 0.006$ | $0.337 \pm 0.009$ | – |
| **PSNR [dB]** ↑ | MNIST | $\mathbf{42.776 \pm 0.018}$ | 3.318 | 5.923 | 0.423 | 1.281 | 1.004 | 3.947 |
| | CelebA | $\mathbf{36.547 \pm 0.113}$ | $5.247 \pm 0.115$ | $6.105 \pm 0.133$ | $4.742 \pm 0.100$ | $4.839 \pm 0.106$ | $4.786 \pm 0.108$ | $5.417 \pm 0.120$ |
| | CelebA-HQ | $\mathbf{40.500 \pm 0.085}$ | $5.349 \pm 0.089$ | – | $4.842 \pm 0.078$ | $4.998 \pm 0.083$ | $4.985 \pm 0.082$ | – |
| **SSIM** ↑ | MNIST | **0.998** | 0.270 | 0.495 | 0.098 | 0.077 | 0.038 | 0.357 |
| | CelebA | **0.991** | $0.051 \pm 0.002$ | $0.153 \pm 0.005$ | $0.100 \pm 0.002$ | $0.015 \pm 0.003$ | $0.007 \pm 0.001$ | $0.081 \pm 0.003$ |
| | CelebA-HQ | **0.991** | $0.072 \pm 0.004$ | – | 0.006 | $0.031 \pm 0.002$ | $0.026 \pm 0.003$ | – |
| **Runtime [sec]** ↓ | MNIST | $\mathbf{0.039 \pm 0.009}$ | $0.096 \pm 0.026$ | $0.085 \pm 0.031$ | $0.098 \pm 0.033$ | $0.101 \pm 0.023$ | $0.092 \pm 0.002$ | $0.122 \pm 0.005$ |
| | CelebA | $\mathbf{0.091 \pm 0.016}$ | $0.109 \pm 0.025$ | $23.956 \pm 0.783$ | $0.112 \pm 0.037$ | $0.118 \pm 0.025$ | $0.111 \pm 0.002$ | $0.592 \pm 0.029$ |
| | CelebA-HQ | $\mathbf{1.258 \pm 0.090}$ | $264.953 \pm 0.043$ | – | $264.849 \pm 0.032$ | $265.016 \pm 0.008$ | $265.019 \pm 0.002$ | – |
| **Memory [MB]** ↓ | MNIST | **0** | **0** | **0** | **0** | **0** | **0** | **0** |
| | CelebA | $0.100 \pm 0.316$ | **0** | **0** | **0** | **0** | **0** | **0** |
| | CelebA-HQ | **0** | 19.981 | – | 19.981 | 19.981 | 19.981 | – |

Table 2: Comparison of STRIP versus iterative methods across datasets. Each method is evaluated at the point where it matches STRIP's single iteration PSNR performance. Best in **bold**.

| Method | MNIST | | | CelebA | | | CelebA-HQ | | |
|---|---|---|---|---|---|---|---|---|---|
| | Iter | Runtime [sec] | Memory [MB] | Iter | Runtime [sec] | Memory [MB] | Iter | Runtime [sec] | Memory [MB] |
| STRIP (Ours) | 1 | $\mathbf{0.039 \pm 0.009}$ | **0** | 1 | $\mathbf{0.091 \pm 0.011}$ | $0.100 \pm 0.316$ | 1 | $\mathbf{1.258 \pm 0.090}$ | **0** |
| CG | 22 | $0.114 \pm 0.034$ | **0** | 50 | $0.194 \pm 0.027$ | **0** | 32 | $265.981 \pm 0.043$ | 19.981 |
| GD | 95 | $0.370 \pm 0.018$ | **0** | 125 | $0.627 \pm 0.048$ | **0** | 900 | $356.911 \pm 0.733$ | 19.981 |
| LSMR | 215 | $0.729 \pm 0.087$ | **0** | 2100 | $48.773 \pm 0.062$ | $0.184 \pm 0.390$ | 555 | $312.230 \pm 0.729$ | 19.981 |
| RSHK | 90 | $4.333 \pm 0.577$ | **0** | 750 | $314.136 \pm 3.557$ | **0** | – | – | – |

inversion method to the noisy inputs. 4. Compare recovered outputs to the analytic PINV solution (or to the original images for CelebA-HQ when PINV is intractable).

**Default Parameters.** Unless noted otherwise: ConvNet $C_{\text{out}} = 8$; Additive noise $1\%$ of input std.; Single iteration; Strip size $h = r$, with $h_b = r/C_{\text{out}}$ rounded to satisfy all the required constraints. We used a batch size of 200 images, and each reported score is the mean and standard deviation computed over 10 independent runs.

## 6.1 IDEMPOTENCY CHECK

We aim to evaluate the compliance of our method with the idempotency measure introduced in Equation (15), expecting potentially lower values with respect to different division methods. The results on CelebA are shown in Figure 5, and the results on MNIST are shown in Figure 11.

## 6.2 COMPARISON WITH ITERATIVE METHODS

We wanted to evaluate our performance versus other iterative methods. The other methods are more general, and are applicable for any linear equation system, not designed to convolution layers like STRIP. For algorithms with native sparse matrix support - *Conjugate Gradients* (CG), *Gauss Seidel* (GS), *Gradient Descent* (GD), *LSQR*, and *LSMR* - we implemented functions that accept sparse matrices as input and operate directly on PyTorch tensors. We also tested *block Kaczmarz* (BK) variants - GBK (Niu & Zheng, 2020), RSHK (Wang & Yin, 2023), RSHEK (Zhang et al., 2024), ADBK (Tan et al., 2025), and FDBK (Chen & Huang, 2022). Because our BK implementations operate on explicit convolution matrices, they are feasible only in lower-dimensional settings. RSHK consistently outperformed the other BK variants, so we include only RSHK in the primary comparisons.

Table 3: Comparison of STRIP versus UPCONV. Best in **bold**.

| Method | MSE ↓ | PSNR [dB] ↑ | SSIM ↑ | Runtime [sec] ↓ | Memory [MB] ↓ |
|---|---|---|---|---|---|
| STRIP (Ours) | **0.0001** | $\mathbf{40.5512 \pm 0.0296}$ | **0.9916** | $1.8275 \pm 0.0980$ | **0** |
| UPCONV | 0.0034 | $25.2383 \pm 0.0001$ | 0.9533 | $\mathbf{0.0033 \pm 0.0061}$ | **0** |

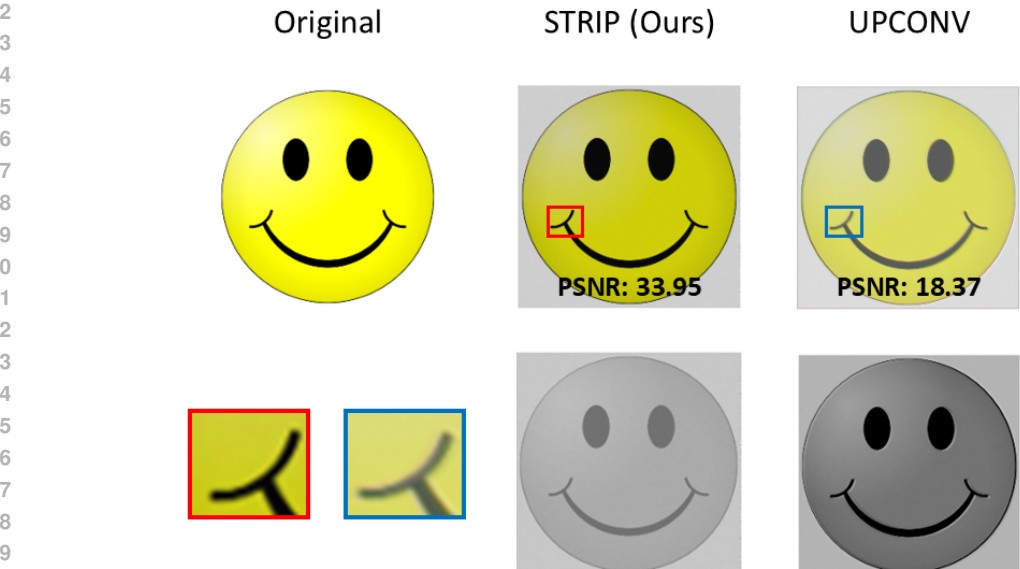

Figure 6: **STRIP vs. UPCONV**: Reconstruction comparison on a synthetic smiley. Top: outputs and PSNR; bottom: difference images (reconstruction - PINV). Zoom-ins show UPCONV suffers from edge blurring, while STRIP preserves sharper boundaries.

In Table 1, our method consistently surpasses all baselines after a single iteration, yielding lower error, faster runtime, and minimal memory usage. For sparse methods, the main cost arises from converting convolution kernels into sparse matrices, with CelebA-HQ exhibiting the sharpest increase due to larger image sizes. Computing the analytic PINV is even more demanding - for example, CelebA requires $2.363 \pm 0.105$ [sec] and $1441.633 \pm 0.575$ [GB]. As shown in Table 2, other methods need many iterations to match STRIP's one-step performance, gaining significant runtime overhead. Results missing from CelebA-HQ correspond to methods infeasible within reasonable time. We denote $0.000 \pm 0.000$ as 0.

### 6.3 COMPARISON WITH TRANSPOSED CONVOLUTION

We benchmark STRIP against a trained UPCONV inverter on CelebA-HQ test set (Table 3). STRIP consistently achieves lower reconstruction error. UPCONV errors are primarily localized around sharp boundaries, often leading to blurred reconstructions, as illustrated in Figure 6 (an OOD example not included in the training or test sets) and in the difference image of Figure 12. While UPCONV also produces edge artifacts, STRIP consistently outperforms it even when such boundary effects are mitigated, demonstrating that STRIP's advantage extends beyond the correction of edge artifacts. While UPCONV offers much faster inference, it requires costly retraining for each dataset, whereas STRIP is training free and directly applicable.

## 7 DISCUSSION AND CONCLUSION

This paper introduced STRIP, an iterative algorithm for convolutional layer inversion. Our method leverages the structure of the convolutional layer matrix by partitioning it into strips in a way that promotes their inter orthogonality. The strips are inverted independently without constructing the full matrix. This design yields reconstructions that are both accurate and efficient, significantly reducing time and memory requirements compared to conventional approaches.

### IMPACT STATEMENT

The superior reconstruction achieved by our method with respect to transposed convolutions, combined with the fact that it is not a learnable layer and therefore requires no retraining across networks,

suggests that it can be deployed in practical scenarios. For example, it can serve as a replacement for up-convolutions in architectures such as U-Net.

## REPRODUCIBILITY

We provide detailed descriptions of theoretical assumptions, proofs, and experimental protocols. Datasets (MNIST (LeCun et al., 1998), CelebA (Liu et al., 2015), and CelebA-HQ (Karras et al., 2018)) are publicly available. Architectures, hyperparameters, and training settings are fully specified (Section 6, Appendix B), and code for experiments will be released to ensure reproducibility.

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

# A    PROOFS AND ADDITIONAL THEORETICAL BACKGROUND

## A.1    FREQUENCY-DOMAIN METHODS

Classical frequency domain deconvolution methods, such as FFT based inverse filtering and Wiener deconvolution, are fundamentally ill suited for inverting multichannel convolutional layers in neural networks. These approaches rely on the assumption of a scalar convolution model, where deconvolution reduces to elementwise division in the frequency domain. In contrast, convolutional layers implement tensor contractions in which each output channel aggregates information from all input channels. This interchannel coupling cannot be decomposed into independent per frequency operations, violating the core premise of classical methods.

The mismatch becomes even more pronounced in the overdetermined setting ($C_{\text{out}} > C_{\text{in}}$), where spatial locality and the inherent sparsity of convolutional kernels are destroyed in the frequency domain. The resulting systems are often ill conditioned, leading to substantial noise amplification. Furthermore, FFT and Wiener based methods neglect the non Gaussian statistics induced by nonlinear activations and the data dependent nature of learned feature representations.

As a result, while mathematically tractable in simplified scenarios, classical frequency domain approaches fail to exploit the structural properties of CNN transformations and provide limited practical utility for accurate or efficient inversion.

## A.2    CONVOLUTION LAYER AS A LINEAR EQUATION SYSTEM

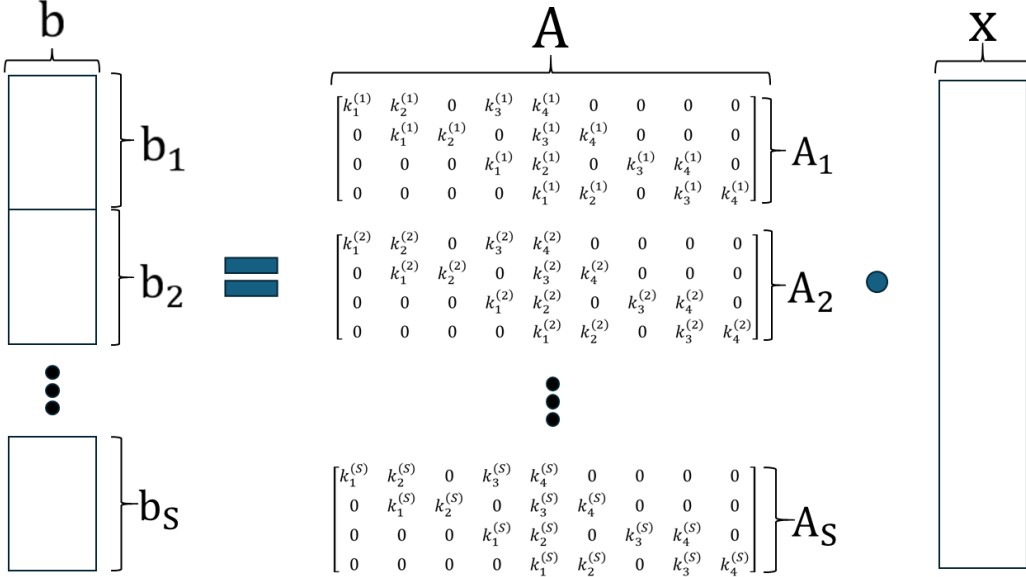

Figure 7: The structure of the convolution matrix, for the simple case of one input channel (grayscale image). Each kernel/output channel corresponds to different block $A_i$ of the full matrix $A$, and effects only the output elements of $b_i$. For illustration, the division into strips is *standard*, where each strip corresponds to the convolution matrix of one kernel, as demonstrated in Figure 4. In practice, however, strips can be organized in more general ways, potentially combining noncontiguous rows or rows belonging to different blocks.

# B    EXPERIMENTS ADDITIONAL MATERIAL

We use a lightweight convolutional network consisting of a single convolution layer with $3 \times 3$ kernels (stride 1), followed by a LeakyReLU activation (slope = 0.5), a $2 \times 2$ average pooling layer, and a fully connected linear layer. The input image of size $(H_{in}, W_{in})$ is reduced to feature maps of size $((H_{in}/2) - 1) \times ((W_{in}/2) - 1)$, which are then flattened and mapped to the output classes.

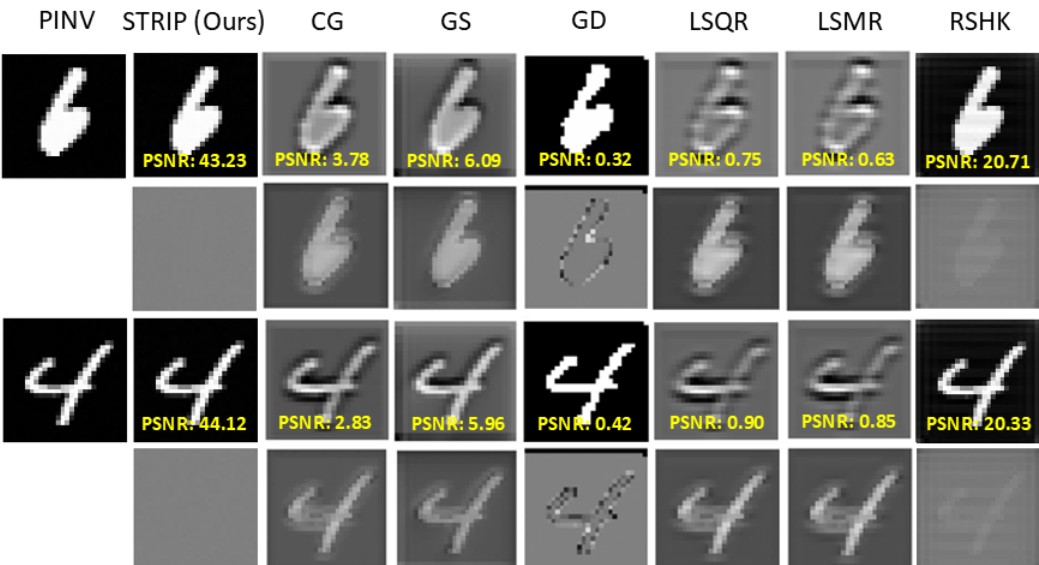

Figure 8: **MNIST deconvolution after a single iteration - 8 convolution channels.** Leftmost column: PINV reference. Odd rows show reconstructions; even rows show error maps (result - PINV). One iteration of our method is a full sweep of the convolution matrix (31 substeps), thus comparable to 31 iterations of the other *block Kaczmarz* baselines.

For all iterative methods, the initialization was set to $x^{(0)} = 0$. GD was implemented with the Adam optimizer (Kingma & Ba, 2015), using a learning rate of $10^{-2}$ in Table 1 and $10^{-1}$ in Tables 2 and 4. GD, which requires a square system, was applied to the normal equations by solving $A^\top A x = A^\top b$ with inputs $A^\top A$ and $A^\top b$, equivalent to Eq. (1). Memory consumption was measured with the `memory_usage` function from the `memory_profiler` package, reporting the difference between maximum and minimum values as the incremental peak memory footprint in megabytes (MB).

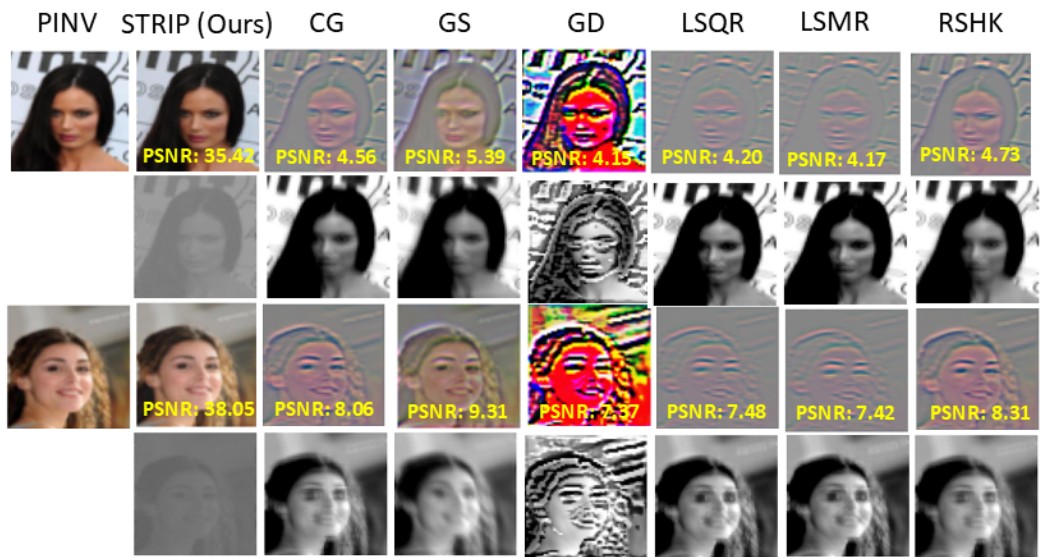

Figure 9: **CELEBA deconvolution after a single iteration - 8 convolution channels.** Leftmost column: PINV reference. Odd rows show reconstructions; even rows show error maps (result - PINV). One iteration of our method is a full sweep of the convolution matrix (31 substeps), thus comparable to 31 iterations of the other *block Kaczmarz* baselines.

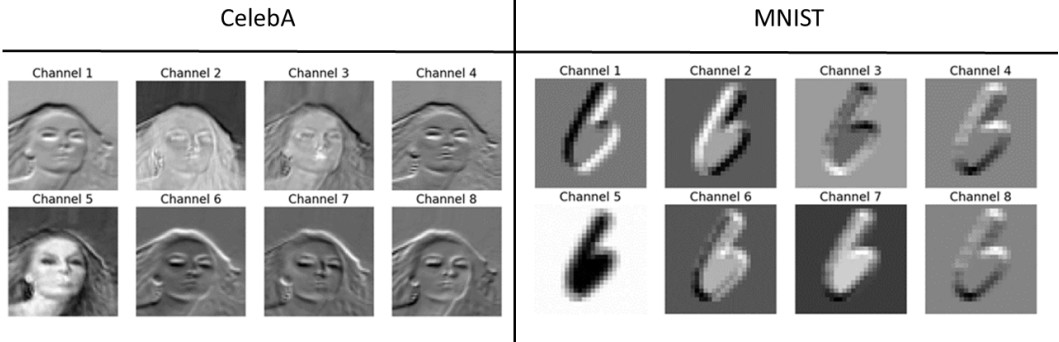

Figure 10: Output channels of convolution result $b$

| | STRIP (Ours) | Interleaved | Standard | Randomized |
|---|---|---|---|---|
| reconstructions | | | | |
| PSNR [dB] | 43.78 | 40.29 | 35.35 | 15.49 |
| Idempotency | 0.001 | 0.001 | 0.495 | 0.653 |

Figure 11: Idempotency of strip arrangements measured by $\|\mathcal{K} - \mathcal{K}^2\|_F$ (lower is better) on MNIST dataset. Smaller values indicate greater inter-strip orthogonality and faster convergence toward the PINV solution.

## C  EXTRA RESULTS

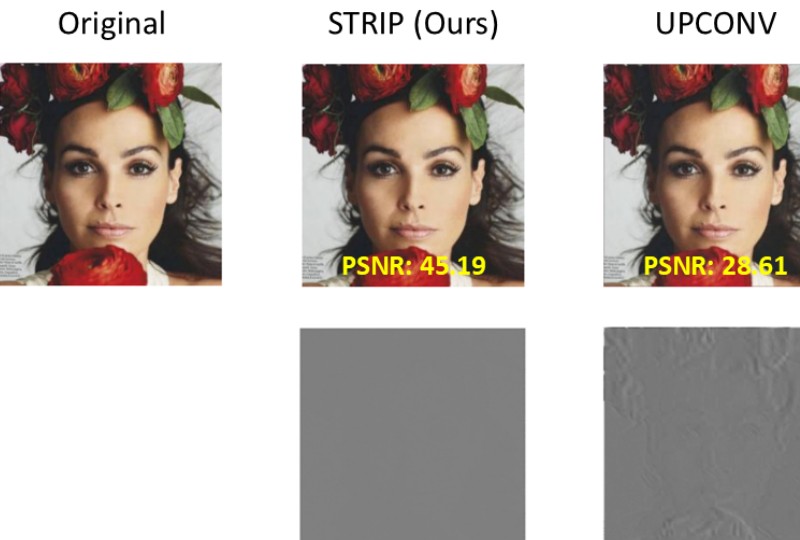

Figure 12: **STRIP vs. UPCONV**: Comparison of reconstruction quality between the two methods. The sample is from the CelebA-HQ dataset. For each case, the first row shows reconstructions and the second row shows difference images (reconstruction - PINV).

## D  JUSTIFICATION OF EQ. (11)

Equation 11 is based on the SVD of the matrices $A_i$

$$A_i = U_i \Sigma_i V_i^T, \quad A_i^\dagger = V_i \Sigma_i^{-1} U_i^{-1}, \quad A_i^\dagger A_i = V_i V_i^T. \tag{16}$$

Table 4: Comparison of STRIP versus iterative methods across datasets with $10\%$ noise. All methods are evaluated after a single iteration. Best in **bold**.

| | | OURS | CG | GS | GD | LSQR | LSMR | RSHK |
|---|---|---|---|---|---|---|---|---|
| **MSE** ↓ | MNIST | **0.005** | 0.470 | 0.258 | 0.764 | 0.747 | 0.797 | 0.408 |
| | CELEBA | **0.024 ± 0.001** | 0.326 ± 0.008 | 0.274 ± 0.007 | 0.327 ± 0.007 | 0.354 ± 0.008 | 0.357 ± 0.008 | 0.314 ± 0.008 |
| | CELEBA-HQ | **0.009** | 0.312 ± 0.006 | – | 0.347 ± 0.005 | 0.336 ± 0.006 | 0.337 ± 0.006 | – |
| **PSNR [dB]** ↑ | MNIST | **23.046 ± 0.023** | 3.303 ± 0.001 | 5.883 ± 0.002 | 1.169 ± 0.001 | 1.270 ± 0.001 | 0.994 ± 0.001 | 3.923 ± 0.001 |
| | CELEBA | **16.571 ± 0.112** | 5.230 ± 0.114 | 5.886 ± 0.133 | 5.185 ± 0.114 | 4.823 ± 0.107 | 4.771 ± 0.105 | 5.398 ± 0.119 |
| | CELEBA-HQ | **20.486 ± 0.081** | 5.348 ± 0.089 | – | 4.841 ± 0.077 | 4.996 ± 0.083 | 4.983 ± 0.083 | – |
| **SSIM** ↑ | MNIST | **0.927** | 0.260 | 0.486 | 0.049 | 0.076 | 0.037 | 0.347 |
| | CELEBA | **0.746 ± 0.003** | 0.047 ± 0.002 | 0.134 ± 0.005 | 0.036 ± 0.001 | 0.014 ± 0.003 | 0.006 ± 0.001 | 0.077 ± 0.003 |
| | CELEBA-HQ | **0.694 ± 0.005** | 0.070 ± 0.003 | – | 0.005 | 0.029 ± 0.002 | 0.025 ± 0.002 | – |
| **Runtime [sec]** ↓ | MNIST | **0.059 ± 0.071** | 0.110 ± 0.065 | 0.803 ± 0.056 | 0.107 ± 0.049 | 0.103 ± 0.033 | 0.094 ± 0.008 | 0.118 ± 0.011 |
| | CELEBA | **0.113 ± 0.098** | 0.125 ± 0.055 | 3.420 ± 0.058 | 3.192 ± 0.051 | 0.126 ± 0.033 | 0.124 ± 0.012 | 0.742 ± 0.021 |
| | CELEBA-HQ | **1.641 ± 0.033** | 605.557 ± 0.054 | – | 605.167 ± 0.045 | 605.585 ± 0.058 | 605.618 ± 0.047 | – |
| **Memory [MB]** ↓ | MNIST | **0** | **0** | **0** | **0** | **0** | **0** | **0** |
| | CELEBA | 0.086 ± 0.273 | **0** | 4.284 ± 4.259 | **0** | **0** | **0** | **0** |
| | CELEBA-HQ | **0.006** | 19.981 | – | 19.981 | 19.981 | 19.981 | – |

Plugging these identities to Equation 9, we get

$$A_i = U_i \Sigma_i V_i^T$$
$$A_i^\dagger = A_i^T \left( A_i A_i^T \right)^{-1}$$

$$= \left( U_i \Sigma_i V_i^T \right)^T \left( U_i \Sigma_i V_i^T \left( U_i \Sigma_i V_i^T \right)^T \right)^{-1} = V_i \Sigma_i U_i^T \left( U_i \Sigma_i \underbrace{V_i^T V_i}_{-I} \Sigma_i U_i^T \right)^{-1} \quad (17)$$

$$= V_i \Sigma_i \underbrace{U_i^T \left( U_i^T \right)^{-1}}_{=I} \Sigma_i^{-2} U_i^{-1} = V_i \Sigma_i^{-1} U_i^{-1}$$

$$A_i^\dagger A_i = V_i \Sigma_i^{-1} U_i^{-1} U_i \Sigma_i V_i^T$$
$$= V_i V_i^T$$

## E  CONVERGENCE IN ONE ITERATION – ANALYSIS

BK recurrence relation in Eq. (12) has the form of a linear difference equation with constant constraint. Given an initial condition $x_0^{(0)}$, the solution to is

$$x_0^{(t)} = \mathcal{K}^t x_0^{(0)} + \left[ I + \mathcal{K} + \cdots + \mathcal{K}^{t-1} \right] \mathcal{B}. \quad (18)$$

The convergence of this solution depends on the spectrum of $\mathcal{K}$ and the respective eigenvectors. Based on Definition 1 in Perko (2012) p. 51, we define here the stable, unstable, and center subspaces of the dynamical system in Eq. (12).

Let $w_j$ be a generalized eigenvector associated with the eigenvalue $\mu_j$ of the matrix $\mathcal{K}$. The stable, center, and unstable subspaces, denoted as $E^s$, $E^c$, and $E^u$, are defined as follows:

$$E^s = \text{Span}\{\text{Re}\{w_j\}, \text{Im}\{w_j\} \,|\, |\mu_j| < 1\} \quad (19)$$
$$E^c = \text{Span}\{\text{Re}\{w_j\}, \text{Im}\{w_j\} \,|\, |\mu_j| = 1\} \quad (20)$$
$$E^u = \text{Span}\{\text{Re}\{w_j\}, \text{Im}\{w_j\} \,|\, |\mu_j| > 1\} \quad (21)$$

The solution ,Eq. (14), converges if $\mathcal{B}$. The constraint on $x_0^{(0)}$ is mitigated. It should also belong to $E^s$ but can have components of eigenvectors with eigenvalue $\mu = 1$. The initial condition is chosen to be zero, then the only concern is whether $\mathcal{B}$ is in $E^s$. Moreover, the fastest convergence occurs when $\mathcal{B}$ belongs to the kernel of $\mathcal{K}$. Let us examine the $i$th row in Eq. (11), given by,

$$x_i^{(t)} = K_i x_{i-1}^{(t)} + V_i \hat{b}_i. \quad (22)$$

where $K_i = I - V_i V_i^T$. The matrix $V_i$ results from the SVD of $A_i$, and therefore $V_i^T V_i = I$. Consequently, the matrix $K_i$ is an idempotent matrix, i.e. $K_i^2 = K_i$. Its eigenvalues are either 0 or 1 associated with columns of $V_i$ and the complementary space, respectively.

If Eq. (22) is the only one in the equation system, we can formulate its solution according to Eq. (14). The constraint $\mathcal{B}$ is $V_i \hat{b}_i$ and belongs to the kernel of $K_i$. Therefore, the solution where $t$ approaches infinity is

$$x_i^f = \lim_{t \to \infty} x_i^{(t)} = K_i x_i^{(0)} + V_i \hat{b}_i = x_i^{(1)}. \tag{23}$$

Consequently, the solution of each row converges in one step, since $K_i$ is an idempotent matrix and the constraint in each row belongs to the kernel of $K_i$. This leads us to the following lemma.

**Lemma 3** (Convergence in one step). *A dynamical system of the form*

$$y_{k+1} = y_k A + B$$

*converges in one step if $A$ is an idempotent matrix and $B$ belongs to its kernel.*

*Proof.* The solution is given by

$$y_k = A^k y_0 + \left[ I + A + A^2 + \ldots + A^{k-1} \right] . B$$

However, $A^k = A$ for all positive integer $k$ since $A$ is an idempotent matrix and $AB = 0$ since $B$ belongs to the kernel space of $A$. Therefore, $y_k = y_1$. Convergence in one step. □

Now, we generalize this conclusion to $S$ rows.

The main questions are

1. How far $\mathcal{B}$ from the kernel of $\mathcal{K}$? or equivalently, what eigenvalues this constraint invokes in this dynamical system?

2. How far $\mathcal{K}$ from being idempotent?

### E.1 ORTHOGONALITY

Let $\mathcal{V}_i$ be the column space of $V_i$ which is the kernel space of $K_i$. From Eq. (13) the vector $\mathcal{B}$ belongs to the union $\cup_{i=1}^{S} \mathcal{V}_i$. Here, we study the relation between the linear dependencies of these subspaces and the convergence of Eq. (14).

**Lemma 4.** *The solution of Eq. (12) (with the structure dictated by Eq. (13)) converges in one step if $\mathcal{V}_i \perp \mathcal{V}_j, \forall i \neq j$.*

*Proof.* The matrices $\mathcal{K}$ and $\mathcal{B}$ become

$$\mathcal{K} = I - \sum_{i=1}^{S} V_i V_i^T, \qquad \mathcal{B} = \sum_{j=1}^{S} V_j \hat{b}_j. \tag{24}$$

since $\mathcal{V}_i \perp \mathcal{V}_j, \forall i \neq j$. Hence, $\mathcal{K}$ is an idempotent and $\mathcal{B}$ belongs to its kernel space. The dynamics convergence in one step, since the conditions of Lemma 3 hold. □

### E.2 LINEAR DEPENDENCY

Here, the subspaces $\mathcal{V}_i$ are not necessarily orthogonal. We start our investigation for $S = 2$ and then try to generalize the conclusions to any $S > 2$.

#### E.2.1 CONVERGENCE CONDITION FOR $S = 2$

**Notations for $S = 2$** The matrix $\mathcal{K}$ from Eq. (13) is

$$\begin{aligned} \mathcal{K} &= \left( I - V_2 V_2^T \right) \left( I - V_1 V_1^T \right) \\ &= I - V_2 V_2^T - V_1 V_1^T + V_2 V_2^T V_1 V_1^T. \end{aligned} \tag{25}$$

Let us denote $C_{2,1} = V_2^T V_1$. The $i,j$th entry of that matrix is the cosine of the angle between the $i$th column vector of $V_2$ and the $j$th column vector of $V_1$. If there are $m_1$ columns in $V_1$ and $m_2$ in $V_2$, one can write $C_{2,1}$ as

$$C_{2,1} = \begin{bmatrix} \cos(\theta_{1,1}) & \cos(\theta_{1,2}) & \cdots & \cos(\theta_{1,m_1}) \\ \cos(\theta_{2,1}) & \cos(\theta_{2,2}) & & \cos(\theta_{2,m_1}) \\ \vdots & & & \\ \cos(\theta_{m_2,1}) & \cos(\theta_{m_2,2}) & & \cos(\theta_{m_2,m_1}) \end{bmatrix}. \tag{26}$$

For more compact writing, we denote the $i$th row in that matrix $c_i = [\cos(\theta_{i,1}) \cdots \cos(\theta_{i,m_1})]$ and the matrix as

$$C_{2,1} = \begin{bmatrix} c_1 \\ \vdots \\ c_{m_2} \end{bmatrix}. \tag{27}$$

Note that if we denote $V_1^T V2$ as $C_{1,2}$ then we get $C_{1,2} = C_{2,1}^T$ and the matrix $C_{2,1} C_{1,2}$ is a Gram matrix and can be formulated as

The matrix $C_{2,1} C_{1,2}$ is symmetric and Gram, and based on Eq. (26) it can be formulated as

$$C_{2,1} C_{1,2} = \begin{bmatrix} \|c_1\|^2 & \langle c_1, c_2 \rangle & \cdots & \langle c_1, c_{m_2} \rangle \\ \langle c_1, c_2 \rangle & \|c_2\|^2 & \cdots & \langle c_2, c_{m_2} \rangle \\ \vdots & & & \\ \langle c_1, c_{m_2} \rangle & \langle c_2, c_{m_2} \rangle & \cdots & \|c_{m_2}\|^2 \end{bmatrix} \tag{28}$$

Now, we can rewrite the dynamics

$$\mathcal{K} = I - V_2 V_2^T - V_1 V_1^T + V_2 C_{2,1} V_1^T. \tag{29}$$

**Convergence Analysis for $S = 2$**

ANSWER TO QUESTION 1    "What eigenvalues the constraint $\mathcal{B}$ are invoked in this dynamical system?"

The vector $\mathcal{B}$ can be formulated as $V_1 \alpha_1 + V_2 \alpha_2$ where $\alpha_1$ and $\alpha_2$ are column vectors with the corresponding dimension.

The eigenvalue/vector admits the following equation,

$$\mathcal{K}(V_1 \alpha_1 + V_2 \alpha_2) = -V_1 C_{1,2} \alpha_2 + V_2 C_{2,1} C_{1,2} \alpha_2 = \lambda(V_1 \alpha_1 + V_2 \alpha_2) \tag{30}$$

where $\lambda \in \mathbb{C}$. By applying the method of variation of parameters, the vectors $\alpha_1$ and $\alpha_2$ admit the following relations

$$\lambda \alpha_2 = C_{2,1} C_{1,2} \alpha_2 \tag{31a}$$

$$\lambda \alpha_1 = -C_{1,2} \alpha_2. \tag{31b}$$

The vector $\alpha_2$ admits the eigenvalue problem of the matrix $C_{2,1} C_{1,2}$. In other words, the eigenproblem of $\mathcal{K}$ leads us to the eigenproblem of matrix $C_{2,1} C_{1,2}$.

To recap, the answer to question 1 is: the spectrum $\mathcal{B}$ invokes is the spectrum of the matrix $C_{2,1} C_{1,2}$.

The upper and lower bounds of the eigenvalues can indicate the system converges and in what minimal pace. The upper bound indicates the slowest pace of convergence of the system, and the lower bound indicates whether the system diverges. We use Gershgorim theorem and on attributes of Gram matrix to find these bounds. Recall the theorem of Gershgorim.

**Theorem 5** (Gershgorin circle theorem Gershgorin (1931)). *Given an $n \times n$ matrix $A$, where $[A]_{i,j} = a_{i,j}$, the eigenvalues are in the following domain in $\mathbb{C}$*

$$\bigcup_{i=1}^{n} B\left(a_{i,i}, \sum_{j=1,j\neq i}^{n} |a_{i,j}|\right) \tag{32}$$

*where $B(a, r)$ is a ball in $\mathbb{C}$ centered in $a$ with radius $r$.*

By applying the Gershgorin theorem to $C_{2,1}C_{1,2}$, the eigenvalues are in the following union $\bigcup_{i=1}^{m_2} B_i$, where

$$B_i = B\left(\|c_i\|^2, \sum_{j=1,j\neq i}^{m_2} |\langle c_i, c_j \rangle|\right). \tag{33}$$

In addition, the matrix in question is Gram. Its spectrum is real and non-negative. Thus, one can reformulate the circle of Gershgorin. The spectrum of $C_{2,1}C_{1,2}$ in the union of the following segments

$$B_i = \left[\max\left\{\|c_i\|^2 - \sum_{j=1,j\neq i}^{m_2} |\langle c_i, c_j \rangle|, 0\right\}, \sum_{j=1}^{m_2} |\langle c_i, c_j \rangle|\right]. \tag{34}$$

Recap: the eigenvalues invoked by $\mathcal{B}$ are the eigenvalues of $C_{2,1}C_{1,2}$ that are contained in $\bigcup_{i=1}^{m_2} B_i$.

**Lemma 6** (Convergence and Divergence for $S = 2$). *A dynamical system of the form of Eq. (29) converges if $I_c < 1$ and diverges if $I_d > 1$, where*

$$
\begin{aligned}
I_c &= \max_{1\leq i\leq m_2}\left\{\sum_{j=1}^{m_2} |\langle c_i, c_j \rangle|\right\} \\
I_d &= \min_{1\leq i\leq m_2}\left\{\|c_i\|^2 - \sum_{j=1,j\neq i}^{m_2} |\langle c_i, c_j \rangle|\right\}
\end{aligned}
\tag{35}
$$

*Proof.* The constraint $\mathcal{B}$ invokes eigenvalues in the union $\bigcup_{i=1}^{m_2} B_i$ where the segment $B_i$ is defined in Eq. (34). The upper bound of this union is

$$I_c = \max_{1\leq i\leq m_2}\left\{\sum_{j=1}^{m_2} |\langle c_i, c_j \rangle|\right\}. \tag{36}$$

Therefore, if the upper bound is less than one, all the eigenvalues are between zero and one. Therefore, convergence.

The lower bound of this union is

$$I_d = \min_{1\leq i\leq m_2}\left\{\|c_i\|^2 - \sum_{j=1,j\neq i}^{m_2} |\langle c_i, c_j \rangle|\right\}. \tag{37}$$

If $I_d$ is larger than one all the eigenvalues are larger than one. Therefore, divergence. $\qquad\square$

The condition $I_c < 1$ does not guarantee only convergence, but also the bound to the slowest the pace of convergence. Therefore, the lower $I_c$, the faster convergence. When $I_c = 0$ the convergence is the fastest and the system gets its steady state in one step. In that case, the matrix $\mathcal{K}$ is idempotent as discussed in the orthogonal case (Lemma 4).

ANSWER TO QUESTION 2   "How far $\mathcal{K}$ from being idempotent?"

The attribute of idempotency is crucial for convergence in one step, as discussed in Lemma 3. Orthogonality and idempotency dictate a special structure for the matrix $\mathcal{K}$, see Lemma 4. Then, our suggestion is to measure the distance of $\mathcal{K}$ from idempotency as follows,

$$\left\|\mathcal{K} - \mathcal{K}^{\perp}\right\|_F^2 \tag{38}$$

where $\mathcal{K}$ is the matrix as in Eq. (29) and $\mathcal{K}^{\perp}$ is the matrix $\mathcal{K}$ is if it were an idempotent matrix, meaning

$$\mathcal{K}^{\perp} = I - \sum_{i=1}^{2} V_i V_i^T. \tag{39}$$

Thus, the distance is defined as,

$$\left\|\mathcal{K} - \mathcal{K}^{\perp}\right\|_F^2 \tag{40}$$

**Lemma 7** (Distance from idempotency and convergence pace $S = 2$). *Let the dynamics be with two strips, $S = 2$. If the distance from idempotency is less than one, the dynamics Eq. (12) converges.*

*Proof.*

$$
\begin{aligned}
\left\|\mathcal{K} - \mathcal{K}^\perp\right\|_F^2 &= \left\|\prod_{i=1}^2 \left(I - V_i V_i^T\right) - \left(I - \sum_{i=1}^2 V_i V_i^T\right)\right\|_F^2 \\
&= \left\|I - \sum_{i=1}^2 V_i V_i^T + V_2 C_{2,1} V_1^T - \left(I - \sum_{i=1}^2 V_i V_i^T\right)\right\|_F^2 \\
&= \left\|V_2 C_{2,1} V_1^T\right\|_F^2 = Tr\left\{V_2 C_{2,1} V_1^T \left(V_2 C_{2,1} V_1^T\right)^T\right\}
\end{aligned}
\tag{41}
$$

The matrix $C_{2,1} C_{1,2}$ is diagonalizable with positive eigenvalues. Then, we can write

$$
\left\|\mathcal{K} - \mathcal{K}^\perp\right\|_F^2 = Tr\left\{V_2 \underbrace{U \Lambda U^T}_{=C_{2,1} C_{1,2}} V_2^T\right\}
\tag{42}
$$

where $V_2 U$ is a unitary matrix. Therefore, the trace is

$$
\left\|\mathcal{K} - \mathcal{K}^\perp\right\|_F^2 = \sum_{i=1}^m \lambda_i.
\tag{43}
$$

where $m = \min\{m_1, m_2\}$. Hence, if the sum, $\sum_{i=1}^m \lambda_i$, is less than one, each eigenvalue is less than one. Then, convergence. $\qquad\square$

Thus, the linear dependency affects the convergence rate.

### E.2.2 GENERALIZATION TO $S > 2$

The analysis of Kaczmarz dynamics for $S = 2$ reveals the complexity to derive quantity indications for pace convergence where $S > 2$. Even for $S = 3$, finding the eigenvalues of $\mathcal{K}$ is a challenging task. Therefore, the answer to the Question 1 is not clear.

On the other hand, the generalization of the distance of $\mathcal{K}$ from idempotency is almost obvious. Let us recall the form of the dynamics

$$
\mathcal{K} = \prod_{i=1}^S \left(I - V_i V_i^T\right),
\tag{44}
$$

and the "idempotent" part of this matrix as

$$
\mathcal{K}^\perp = I - \sum_{i=1}^S V_i V_i^T.
\tag{45}
$$

If the subspaces $\{\mathcal{V}_i\}_{i=1}^S$ are orthogonal, the matrix $\mathcal{K}$ would be equal to $\mathcal{K}^\perp$. The other (correlated) addends are for the dependency between these subspaces. The lower the linear dependence, the lower the correlated addends and therefore, the closer $\mathcal{K}$ to $\mathcal{K}^\perp$. The distance from idempotency is the norm of the correlated addends in $\mathcal{K}$, or more formally,

$$
\left\|\prod_{i=1}^S \left(I - V_i V_i^T\right) - \left(I - \sum_{i=1}^S V_i V_i^T\right)\right\|_F^2.
\tag{46}
$$

Let us denote the matrix $\mathcal{K}$ as follows,

$$
\mathcal{K} = \mathcal{K}^\perp + \begin{bmatrix} V_1 & V_2 & \dots & V_S \end{bmatrix} C_{\mathcal{K}} \begin{bmatrix} V_1 & V_2 & \dots & V_S \end{bmatrix}^T
\tag{47}
$$

where the matrix $C_{\mathcal{K}}$ contains all the correlations between all the possible combinations of the products in Eq. (44).

$$
\begin{aligned}
\left\|\mathcal{K} - \mathcal{K}^{\perp}\right\|_F^2 =& Tr\{[V_1 \quad V_2 \quad \dots \quad V_S]\, C_{\mathcal{K}} C_S C_{\mathcal{K}}^T\, [V_1 \quad V_2 \quad \dots \quad V_S]^T\} \\
=& Tr\{C_{\mathcal{K}} C_S C_{\mathcal{K}}^T\, [V_1 \quad V_2 \quad \dots \quad V_S]^T\, [V_1 \quad V_2 \quad \dots \quad V_S]\} \\
=& Tr\{C_{\mathcal{K}} C_S C_{\mathcal{K}}^T C_S\}
\end{aligned}
\tag{48}
$$

where $C_S = [V_1 \quad V_2 \quad \dots \quad V_S]^T\, [V_1 \quad V_2 \quad \dots \quad V_S]$. The distance in the general case is not directly the sum of the eigenvalues of $\mathcal{K}$. However, if $\mathcal{V}_i \perp \mathcal{V}_j$ for all $i \neq j$, the distance from idempotency is zero, and the convergence is achieved in one step.

### E.3 CONVERGENCE FOR $S > 2$

Let us denote the matrix $\mathcal{K}$ as follows,

$$
\mathcal{K} = \mathcal{K}^{\perp} + [V_1 \quad V_2 \quad \dots \quad V_S]\, C_{\mathcal{K}}\, [V_1 \quad V_2 \quad \dots \quad V_S]^T
\tag{49}
$$

where the matrix $C_{\mathcal{K}}$ contains all the correlations between all the possible combinations of the products in $\mathcal{K} = \prod_{i=1}^{S} \left(I - V_i V_i^T\right)$. Therefore, the distance for the general case is as follows,

$$
\begin{aligned}
\left\|\mathcal{K} - \mathcal{K}^{\perp}\right\|_F^2 =& \left\|[V_1 \quad V_2 \quad \dots \quad V_S]\, C_{\mathcal{K}}\, [V_1 \quad V_2 \quad \dots \quad V_S]^T\right\|_F^2 \\
=& Tr\{[V_1 \quad V_2 \quad \dots \quad V_S]\, C_{\mathcal{K}} C_S C_{\mathcal{K}}^T\, [V_1 \quad V_2 \quad \dots \quad V_S]^T\} \\
=& Tr\{C_{\mathcal{K}} C_S C_{\mathcal{K}}^T\, [V_1 \quad V_2 \quad \dots \quad V_S]^T\, [V_1 \quad V_2 \quad \dots \quad V_S]\} \\
=& Tr\{C_{\mathcal{K}} C_S C_{\mathcal{K}}^T C_S\}
\end{aligned}
\tag{50}
$$

where $C_S = [V_1 \quad V_2 \quad \dots \quad V_S]^T\, [V_1 \quad V_2 \quad \dots \quad V_S]$. The distance from idempotency is the norm of the correlated addends in $\mathcal{K}$. The distance in the general case is not directly the sum of the eigenvalues of $\mathcal{K}$ as when $S = 2$ (see Lemma 7). However, if $\mathcal{V}_i \perp \mathcal{V}_j$ for all $i \neq j$, the distance from idempotency is zero, and the convergence is achieved in one step. The lower the linear dependence, the lower the correlated addends and therefore, the closer $\mathcal{K}$ to $\mathcal{K}^{\perp}$. From our experiments, the lower the distance the faster the convergence. In most cases, after one iteration the algorithm gets to its final result. Consequently, the following conjecture is backed up with experiments, however, we did not find the relation of this distance to eigenvalues of the correlation matrix $C_{\mathcal{K}}$ and $C_S$.

**Conjecture 8** (Convergence pace and distance from Idempotency). *Let the distance from idempotency be $\left\|\mathcal{K} - \mathcal{K}^{\perp}\right\|_F^2$. If the distance is zero, the system converges in one step. In addition, the lower the distance, the faster the convergence rate.*

### E.4 PROOF OF THEOREM 2

*Proof.* The proof of Theorem 2 follows from Lemma 3, Lemma 4, Lemma 6, and Lemma 7. $\square$

