# OpenReview forum: "FAST AND SCALABLE INVERSION OF CONVOLUTION LAYERS"
_ICLR.cc/2026/Conference — ICLR 2026 Conference Withdrawn Submission_

### Official Review · Reviewer_NLxa · 2025-10-23

**Soundness:** 2
**Presentation:** 3
**Contribution:** 2
**Rating:** 2
**Confidence:** 3

**Summary:**

This paper introduces STRIP, a method for convolution inversion. The authors formulate convolution as a linear system of equations and employ the block Kaczmarz (BK) algorithm to approximate the solution. The method leverages toeplitz structure of the coefficient matrix in this formulation and proposes an efficient row selection mechanism required by the BK algorithm. Specifically, the authors rearrange the rows of the coefficient matrix so that the convolution kernel repeats in blocks. As a result, the necessary computations for the BK algorithm can be performed on a single block and then reused across the others. The paper analyzes this mechanism in the context of the BK algorithm and compares it against standard iterative baselines.

**Strengths:**

- The paper is overall easy to follow, and the proposed row selection mechanism for the BK algorithm is novel.

- The use of the BK algorithm for convolution inversion is, to my knowledge, new.

- The theoretical analysis is thorough, and the proposed method is shown to be computationally efficient under certain conditions.

**Weaknesses:**

- The method is not well motivated, and its practical application is unclear. The authors argue that it can be used in U-Nets; however, they do not showcase such a use case in their experiments. Therefore, its application appears to be limited to computing convolution inverses in non-learning scenarios, without offering clear practical benefits for machine learning domain.

- The paper’s main contribution, the select_rows function in Algorithm 1, is only described intuitively in the text. It would be clearer and more useful if this mechanism were presented as pseudocode.

- While the mathematical analysis of the method is solid, it does not provide much insight for practical use. In particular, the connection between the results in Theorem 2 and the convolution kernel or its parameters is non-intuitive. What is the key takeaway from Theorem 2, and what are its implications for using the proposed method?

- I am not familiar with existing works on convolution inversion; however, the baselines used in the experiments do not appear to be designed specifically for convolutions. Therefore, it is not surprising that these baselines perform much worse, as shown in Tables 1 and 2. The authors should include related baselines, and if no such methods exist, at least provide a naive version of the BK algorithm: how do the other row arrangements shown in Figure 4 perform?

- The experiment comparing the method with transposed convolution is not clearly described and is very limited.

- The writing also needs improvement and is somewhat messy in its current form. For example, some figures are not referenced in the text, section 5 could be compressed by moving unnecessary details to the appendix, and the numbering of elements (theorem, figures, etc) needs to be fixed.

**Questions:**

Please see weaknesses.

---

### Official Review · Reviewer_ah6f · 2025-10-30

**Soundness:** 2
**Presentation:** 2
**Contribution:** 1
**Rating:** 2
**Confidence:** 4

**Summary:**

The authors propose a novel method for inversion of convolutional layers. Their method is augmented variant of the block Kaczmarz algorithm, which exploits the structure of convolutional layers. The authors showcase the application of their method on a variety of different datasets.

**Strengths:**

The authors showcase that their method significantly improves on computation time as well as PSNR and MSE. The authors evaluated this work on multiple datasets.

**Weaknesses:**

1. Limited applicability. The authors demonstrate their method in a highly artificial and non-practical context. The inversion of a single convolutional layer is not a realistic or meaningful application. The paper does not present any practical use cases, only citing the U-Net architecture, which uses learnable transposed convolutions. The closest real-world scenario involving convolutional inverses is in normalizing flows; however, even there, strict constraints are imposed to ensure the Jacobian determinant can be computed efficiently.

2. Limited experiments. The authors conduct experiments using only 8 or 16 channels, whereas typical architectures employ 256–512 channels. Under such realistic conditions, the proposed method would likely be computationally infeasible.

3. Computation time. Although the authors report that their method takes 1.8275 seconds compared to 0.0033 seconds for an up-convolution (upconv), this makes the approach practically unusable. Such computational overhead would become a major bottleneck in any real-world setting.

4. Incomplete comparison with upconv. The authors claim higher PSNR compared to upconv but provide no details regarding the experimental setup. The main advantage of upconv is that it can be trained—so why not train it? Running even a few training iterations (within the same 2-second budget) could yield better results while also producing a faster inverse operation.

5. Memory experiments. Almost all experiments showcase 0 memory consumption, which is doubtfully true.

**Questions:**

1. Could you comment on the weaknesses that I pointed?
2. What is the main potential application of the method?
3. What are the matrix constraints we need to impose in order to have a stable inverse? During training a NN weights of the conv layers shift, and the condition number can change, and the inverse might be infeasible due to aggregate numerical error.

---

### Official Review · Reviewer_4pZV · 2025-11-01

**Soundness:** 3
**Presentation:** 3
**Contribution:** 2
**Rating:** 6
**Confidence:** 4

**Summary:**

The paper targets the inversion of the convolution layer (linear). It differs from the learning-based inverse to derive mathematical analysis on the convolutional layer matrices. The proposed algorithm is related to the classic pseudoinverse, but in a partitioned manner to have fast computation and convergence.

**Strengths:**

The problem scope is made clear, allowing for a fast inverse computation for convolutional layers without additional training.

The partitioning idea based on pseudoinverses to reduce memory and computation sounds like the right track and is easy to follow.

The experiments comparing the proposed method with UPCONV and other solvers show an obvious improvement.

**Weaknesses:**

For the target problem, the proposed method seems to apply only to a per-layer linear inverse. Is it true? Or are the compositions across layers considered? If stacking is intended, what breaks or changes (conditioning, caching pseudoinverse of A, stability)?

As images are mainly used, a natural question is whether the convolution inversion is trivial or ill-conditioned. Conceptually, images may contain information redundancy, and convolution layers are often used to compress and extract information. In contrast, inverse operations need to preserve information without any loss, as it is trivial to trace back to the origin. The paper also argues classic deconvolution is ill-conditioned and mismatched to multi-channel conv. Is it possible to quantify the conditioning of the actual convolution operators for inversion?

What is the use case of the analyzed linear inverse, e.g., to replace transposed conv in decoders? While the Related Work mentions that learning an additional inverse model lacks explicitness and is computationally heavy, the reviewer wonders about the performance comparison of the proposed analytical inversion with inverse modeling. Inverse modeling also includes analytically invertible models and numerical inverse approximation. Will the proposed method be more accurate?

**Questions:**

Please see my concerns and questions in Weaknesses.

---

### Note · Authors · 2025-11-14

I have read and agree with the venue's withdrawal policy on behalf of myself and my co-authors.